# Spatial-Temporal Evolution and Driving Forces of NDVI in China’s Giant Panda National Park

**DOI:** 10.3390/ijerph19116722

**Published:** 2022-05-31

**Authors:** Mengxin Pu, Yinbing Zhao, Zhongyun Ni, Zhongliang Huang, Wanlan Peng, Yi Zhou, Jingjing Liu, Yingru Gong

**Affiliations:** 1College of Earth Sciences, Chengdu University of Technology, Chengdu 610059, China; pumengxin@163.com (M.P.); nizhongyun2012@mail.cdut.edu.cn (Z.N.); hzl19961002@gmail.com (Z.H.); 2College of Tourism and Urban-Rural Planning, Chengdu University of Technology, Chengdu 610059, China; pengwanlan@stu.cdut.edu.cn (W.P.); zhouyi1@stu.cdut.edu.cn (Y.Z.); lj0011822@gmail.com (J.L.); yr852587432@163.com (Y.G.); 3Hebei Province Key Laboratory of Wetland Ecology and Conservation, Hengshui University, Hengshui 053000, China; 4International Network for Environment and Health, School of Geography and Archaeology & Ryan Institute, National University of Ireland, Galway (NUIG), H91 CF50 Galway, Ireland

**Keywords:** vegetation, geographically weighted regression, climate change, Wenchuan earthquake, ecological security

## Abstract

Identifying the ecological evolution trends and vegetation driving mechanisms of giant panda national parks can help to improve the protection of giant panda habitats. Based on the research background of different geomorphological zoning, we selected the MODIS NDVI data from 2000 to 2020 to analyze the NDVI trends using a univariate linear model. A partial correlation analysis and multiple correlation analysis were used to reveal the influence of temperature and precipitation on NDVI trends. Fourteen factors related to meteorological factors, topographic factors, geological activities, and human activities were selected, and the Geographically Weighted Regression model was used to study the mechanisms driving NDVI change. The results were as follows: (1) The NDVI value of Giant Panda National Park has fluctuated and increased in the past 21 years, with an annual growth rate of 4.7%/yr. Affected by the Wenchuan earthquake in 2008, the NDVI value fluctuated greatly from 2008 to 2012, and reached its peak in 2018. (2) The NDVI in 94% of the study area improved, and the most significant improvement areas were mainly distributed in the northern and southern regions of Southwest Subalpine and Middle Mountain and the Xiaoxiangling area. Affected by the distribution of fault zones and their local activities, vegetation degradation was concentrated in the Dujiangyan–Anzhou area of Hengduan Mountain Alpine Canyon. (3) The Geographically Weighted Regression analysis showed that natural factors were dominant, with climate and elevation having a double-factor enhancement effect, the peak acceleration of ground motion and fault zone having a superimposed effect, and river density and slope having a double effect, all of which had a significant impact on the NDVI value of the surrounding area. To optimize the ecological security pattern of the Giant Panda National Park, we recommended strengthening the construction of ecological security projects through monitoring meteorological changes, preventing, and controlling geo-hazards, and optimizing the layout and intensity of human activities.

## 1. Introduction

The giant panda is an important national protected animal in China, and changes in its habitat quality have received extensive attention from researchers. Affected by urban and rural construction activities, the fragmentation of giant panda habitats is serious, and the natural and anthropogenic threats to the habitat quality continue to intensify [1,2,3]. National parks are key areas for strictly protecting biodiversity in various countries [4], with the purpose of achieving effective ecological protection, rational utilization of resources and sustainable social and economic development [5,6]. In order to adapt to the multi-departmental collaborative management of giant panda reserves, the Chinese government proposed the establishment of the China Giant Panda National Park (CGPNP) in 2017 and announced its formal establishment in 2021 [7]; this balances the needs of regional ecological protection and socio-economic development and continuously improves the ability of biodiversity protection to balance the needs of regional ecological protection and socio-economic conservation [8,9]. Vegetation in CGPNP is affected by the complex interaction of ecological elements such as soil, atmosphere, and water, and is also an important medium for natural ecosystems and human production activities [10]. Among many types of remote sensing data, the Normalized Difference Vegetation Index (NDVI) is a sensitive parameter of surface vegetation coverage and vegetation growth which reflects the difference between the radiation absorption in the red spectral region caused by chlorophyll and the reflectivity of canopy structure caused by the NIR spectral region, and it can effectively characterize the vegetation environment and its changes and effects [11,12]. The NDVI generated from remote sensing data have the advantage of a long time series, wide coverage, and high spatial resolution [13], and there are many cases of NDVI being used to monitor national park vegetation, ecological environments and their changes all over the world, including passive monitoring over a long time series [14], vegetation and climate coupling characteristics research [15], vegetation phenology characteristics research [16], and so on.

Many studies have been conducted on vegetation changes in CGPNP reserves in China [17], but these study areas consisted mainly of independent reserves in CGPNP, and the data used are mostly SAR images [18,19,20]. Although the use of SAR image improves the accuracy of image recognition, the coverage area of SAR images is small and the period is limited, so it is not suitable for long-term dynamic monitoring of vegetation. The Moderate Resolution Imaging Spectroradiometer (MODIS) can better solve this problem and can be used for monitoring the vegetation environment and its changes in CGPNP. The research on the trends of changes in vegetation environments is the key component of dynamic monitoring, and the methods involved include the linear regression analysis [21], Mann–Kendall test [22,23], BFAST trend analysis and Theil–Sen median slope trend analysis [24]. Among these, the linear regression analysis is a more effective method in this kind of research.

The driving mechanisms of climate and other factors based on the trends of changes in NDVI form the basis for the formulation of national park protection strategies. Studying the response mechanism of climate change to NDVI changes is of great significance in predicting vegetation dynamics [25]. In the global research on the relationship between vegetation and climate, the climate-driving mechanisms in different regions show significant geographical differentiation [26,27,28,29]. The CGPNP is located in a climate transition zone, and climate drives NDVI with great complexity and uncertainty. Most previous studies on NDVI drivers in this region have only considered climatic factors. For example, Lin et al. focused on the two factors of temperature and precipitation in their research on NDVI-driving forces in North China [30], and Liu et al. also only considered the spatial characteristics of climate factors in the study of vegetation change in China [31]. The impact of geological activities, topography, and human activities was not considered sufficiently [32]. Affected by the 12 May 2008 Wenchuan earthquake and its triggered geo-hazards, the local NDVI decreased rapidly and the habitat quality deteriorated seriously [33,34]. In the more than 10 years since the earthquake, the ecological geological environment has undergone great changes, which has increased the spatial instability and complexity of the analysis of the driving forces of NDVI changes [35]. At present, there are many research methods on NDVI’s driving forces, such as the enhanced regression tree model [36], the Geographically Weighted Regression (GWR) model [37], the correlation analysis method [38] and residual trend analysis [39], most of which are only used for unilateral aspects such as climate factors, ignoring the correlation and coupling between multiple factors and failing to take into account the spatial difference of the action of driving factors. To study the driving mechanisms of NDVI changes, the GWR model is a suitable choice because it enables one to change the parameter settings on the local scale, facilitates the determination of local coefficients, and can solve the problems of spatial instability and scale dependence to a certain extent in the analysis [40,41]. The GWR model can reveal the spatiotemporal variability between each driving factor and vegetation activity by studying the spatial non-stationary relationship between each driving factor and NDVI change value.

Aiming at the complex eco-geological environment of CGPNP, the MODIS NDVI data products were selected, and the Savitzky–Golay filter was used to construct NDVI serial data from 2000 to 2020. In this study, the univariate linear model was used to monitor the interannual NDVI trends in the study area based on geomorphological zoning. The GWR model was used to identify the driving mechanisms of NDVI trends by integrating the effects of natural and human factors. Finally, combined with the temporal and spatial differentiation characteristics and driving mechanisms of NDVI change, eco-geological environmental protection countermeasures were proposed according to the local conditions.

## 2. Study Area and Data Processing

### 2.1. Study Area

The CGPNP is located in the ecological barrier area of the Sichuan–Yunnan Loess Plateau in the “two screens and three belts” ecological barrier area in China’s ecological security strategy [42] with the largest population, protected area type and quantity in China [43]. The establishment of CGPNP brings together various nature reserves and increases the connectivity of giant panda habitats [44,45,46]. From northeast to southwest, the Sichuan Area of China Giant Panda National Park (SCOCGPNP) consists of seven cities (prefectures), including 19 counties (cities and districts). It spans five areas: Qinling Mountains, Baishui River, Minshan Mountain, Qionglai Mountain–Daxiangling, and Xiaoxiangling (Figure 1a). The SCOCGPNP ranges from 102°27′ to 105°57′ E and 29°42′ to 33°34′ N, covering an area of 20,177 km^2^. As the study area is located in the transition area from a subtropical zone to a warm temperate zone (Figure 1b) [47], the average annual precipitation is 830 mm, and the average temperature is 10–17 °C. In the study area, the altitude decreases from northwest to southeast, and the vertical distribution of vegetation is obvious: subtropical evergreen deciduous forest, evergreen deciduous broad-leaved mixed forest, temperate coniferous forest, cold temperate coniferous forest, shrub, and meadow.

In order to fully describe the influence of geomorphology on the basement of NDVI and its changes, the study area was divided into five geomorphological areas (Figure 1c) [48], including the Baishui River–Minshan Area of Southwest Subalpine and Middle Mountain (BSMS), the Minshan Area of Hengduan Mountain Alpine Canyon (MS), the Dujiangyan–Anzhou Area of Hengduan Mountain Alpine Canyon (DA), the Daxiangling Area of Hengduan Mountain Alpine Canyon (DXL), and the Daxiangling-Xiaoxiangling Area of Southwest Subalpine and Middle Mountain (DXLXXL).

### 2.2. Data Sources

#### 2.2.1. MODIS NDVI Data

In this research, the Moderate-resolution Imaging Spectroradiometer (MODIS) spectral imager on the EOS/Terra satellite was used to obtain MOD13A1 products, among which MODIS VI products can be used to monitor the terrestrial photosynthetic vegetation activities of the earth and support the phenology and change monitoring of vegetation in national parks. The NDVI data from 2000 to 2020 in this research were downloaded from the official website of NASA (https://earthdata.nasa.gov/, accessed on 20 February 2021). The product used in this research was the collection 6 data set. Compared with the collection V data set, the 8-day surface reflection data (pre-synthesized based on the Terra and Aqua data) were used, the CV-MVC synthesis method was modified, and the necessary SDS was adjusted to reflect the new input data flow, which improves the change-detection ability of the product [49]. We selected the best available pixel value from the data set collected in 16 days with a resolution of 500 m, resulting in NDVI values with low cloud cover and low viewing angles [50].

#### 2.2.2. Driving Factors

The GWR model was selected to explore the driving mechanism of various factors influencing NDVI trends, and five categories and 14 variables were selected (Table 1).

As the main controlling factor of vegetation structure, composition, and distribution, there is a strong correlation between precipitation and NDVI [51]. Since temperature can regulate the photosynthesis of vegetation [52], annual mean temperature and variability were used as analytical indicators (Figure 2a–d).

Located in a geomorphological boundary area, the study area has complex geomorphology. Among the topographic factors, elevation often determines the temperature and CO_2_ content of vegetation growth (Figure 2e), slope represents soil moisture and sunlight exposure for vegetation growth (Figure 2f), and aspect also has a certain impact on vegetation lighting conditions [53] (Figure 2g).

Located near the Longmenshan fault zone, the crustal activity in the study area affects the growth of vegetation, the looseness of soil, and the stability of its growth state. The Euclidean distance from fault (Figure 2h) and the peak acceleration of ground motion (Figure 2i) were used to characterize the activity of the geological activities [54].

Human activities around construction land, such as logging, grazing, fire, etc., which disturb the original structure of the vegetation landscape, are closely related to changes in the NDVI [55]. In addition, most of the study area is mountainous. Therefore, compared with nighttime light data and POI data, choosing the Euclidean distance for construction land can more intuitively characterize the intensity of human activities, and can exclude errors caused by terrain fluctuations and weak human activities (Figure 2j). The construction and use of roads have a devastating impact on the ecological environment, and the towns and villages connected by them imply the impact of human activities. The denser the road distribution (Figure 2k), the more serious the damage to the vegetation [31]; so the road density was selected as one influencing factor. Due to the large proportion of forest land and mountainous terrain in the study area, the dynamic change in human activities was small, so the static data of 2020 were selected to characterize the impact of human activities.

The distribution of rivers affects the stability of regional NDVI and controls ecological changes and landscape dynamics [56]; therefore, the river density was chosen to characterize its impact (Figure 2l). Different land-use types represent different NDVI values and different NDVI change possibilities. For example, the conversion of construction land or cultivated land to grassland or forest land is beneficial to the improvement of NDVI [57]. In the same geographical environment, the order of NDVI values from high to low is forest land > cultivated land > grassland [58]; whereas a negative value of NDVI usually represents water [59]. Referring to this law, the forest land, cultivated land, grassland, construction land, water bodies, and other land types in the initial and final years were designated as 25, 15, 10, 0, −5, and 0, respectively (Figure 2m). A measure of the transformation of land-use types was obtained by subtracting the initial year from the final year. The NDVI value of the starting year in 2000 was used to characterize the vegetation trends and to assess whether the regional vegetation had reached a saturated state [60] (Figure 2n).

Different driving factors have spatial differences in different partitions (Table A1) and applying this fact to the GWR model is conducive to the discussion of the driving mechanisms of NDVI.

### 2.3. Method

#### 2.3.1. Data Preprocessing

We selected the annual data synthesized from monthly data from 2000 to 2020 as the research data of long time series. There are various synthesis methods available for NDVI data, such as the average value method (AVM), maximum value composite (MVC), time series reconstruction method, and so on. To remove the NDVI outliers in multi-temporal images to a certain extent [61], the MVC method was selected to synthesize the data with a period of 16 days to obtain the monthly NDVI value from 2000 to 2020, so as to eliminate the deviation caused by atmospheric interference, solar elevation, cloud coverage, etc. To eliminate the noise caused by sensor error or cloud cover in data acquisition, Savitzky–Golay filtering was carried out on the NDVI value of continuous time series, and the NDVI value images of 21 consecutive years from 2000 to 2020 were obtained (Figure A1).

#### 2.3.2. NDVI Change Detection

The univariate linear regression analysis model monitors the NDVI time series through the regression time function. The model can calculate the interannual variability of vegetation, which is the slope of the linear regression equation [62]. The calculation formula is as follows:(1)$$\theta_{slope}=\frac{n\times \sum_{i=1}^{n}i\times {\mathrm{NDVI}}_{i}-\sum_{i=1}^{n}i\sum_{i=1}^{n}\;{\mathrm{NDVI}}_{i}}{n\times \sum_{i=1}^{n}i^{2}-{\left(\sum_{i=1}^{n}i\right)}^{2}}$$
where *n* represents the time span and *i* represents the NDVI value in the *i*th year; when $\theta_{slope}\;$> 0, it indicates that NDVI is in an improved state, and when $\theta_{slope}\;$< 0, it indicates that NDVI is degraded.

The *t*-test was used to obtain the significance of the temporal trend of NDVI:(2)$$t=\frac{{\bar{x}}_{1}-{\bar{x}}_{2}}{s\times \sqrt{\frac{1}{n_{1}}+\frac{1}{n_{2}}}}\;$$
(3)$$s=\sqrt{\frac{n_{1}S_{1}^{2}+n_{2}S_{2}^{2}}{n_{1}+n_{2}-2}}$$
where ${\bar{x}}_{1}$ and ${\bar{x}}_{2}$ represent the mean of the two subsamples, *n*_1_ and *n*_2_ represent the number of the two subsamples and *S*_1_ and *S*_2_ represent the standard deviation of the two subsamples.

Through the above process, the change in slope of NDVI and *t*-test results were obtained. The $\theta_{slope}$ was divided into improvement and degradation, with the value of 0 as the boundary. Two confidence levels of 0.01 and 0.05 were selected in the *t*-test to evaluate the significance of the change in NDVI. A *p*-value of 0.01 < *p* < 0.05 was considered significant, *p* < 0.01 was extremely significant, and *p* > 0.05 was considered insignificant. The combination of the two can be used to divide the NDVI trend into six categories: extremely significant degradation, significant degradation, insignificant degradation, insignificant improvement, significant improvement, and extremely significant improvement (Table 2).

#### 2.3.3. NDVI Climate-Driven Analysis

In order to describe the correlation between NDVI, precipitation, and temperature, the partial correlation coefficient and multiple correlation coefficient were selected as quantitative indicators to test significance, and the study area was divided into different climate-driving types according to the test results.

A partial correlation analysis can measure the correlation between two factors under the exclusion of other factors [63]. The formula for calculating the partial correlation coefficient between temperature, precipitation, and NDVI is as follows:(4)$$r_{123}=\frac{r_{12}-r_{13}*r_{23}}{\sqrt{\left(1-r_{13}{}^{2}\right)*\left(1-r_{23}{}^{2}\right)}}$$
(5)$$r_{xy}=\frac{\sum_{i=1}^{n}\left(x_{i}-\bar{x}\right)\left(y_{i}-\bar{y}\right)}{\sqrt{\sum_{i=1}^{n}{\left(x_{i}-\bar{x}\right)}^{2}*\sum_{i=1}^{n}{\left(y_{i}-\bar{y}\right)}^{2}}}$$
where $r_{12},\;r_{13},$ and $r_{23}$ are the correlation coefficients between NDVI and temperature, NDVI and precipitation, and temperature and precipitation; $r_{123}$ is the partial correlation coefficient between the two parameters based on the third parameter.

A statistically significant confidence level of 0.05 was selected, and a *t*-test was performed on the results of partial correlation analysis to obtain the significance between climate-driving factors.
(6)$$t=\frac{r_{123}}{\sqrt{1-r_{123}{}^{2}}}\sqrt{n-m-1}$$
where *n* is the number of samples and *m* is the independent variable.

A multiple correlation analysis was used to study the degree of correlation between NDVI and precipitation and temperature, thereby revealing the driving mechanism of climate on NDVI changes.
(7)$$r_{x,yz}=\sqrt{1-\left(1-r^{2}{}_{xy}\right)\left(1-r^{2}{}_{xz,y}\right)}$$

A significant F-test was performed on the results of the multiple correlation analysis:(8)$$F=\frac{r^{2}{}_{x,yz}}{1-r^{2}{}_{x,yz}}\times \frac{n-k-1}{k}$$
where *n* is the number of samples and *k* is the number of independent variables.

Based on the existing precipitation, temperature, and NDVI data, a partial correlation and multiple correlation analysis were carried out to test the correlation of the two coefficients. In the partial correlation analysis, the *t*-test with a confidence level of 0.05 was used to divide the precipitation-driving type and temperature-driving type pixel by pixel. In the multiple correlation analysis, the F-test with a confidence level of 0.01 was used to determine whether it was a climate-driven region. The combination of the two can be used to obtain the climate-driven zoning map in the study area (Table 3).

#### 2.3.4. NDVI-Driving Force Analysis Based on the GWR Model

The GWR model, which is a spatial decomposition of traditional regression models, can be extended via the estimation of local parameters. The parameters of each spatial point in the entire model are independently quantified, which is often used to test the existence of spatial non-stationarity in the relationship between dependent variables and independent variables [64]. The model can be used to characterize the effects of geology, meteorology, and human activities on vegetation coverage at different spatial locations. The GWR technique extends the traditional global regression by adding a geolocation parameter, and the formula is as follows:(9)$$y_{i}=\beta_{o}\;\left(\mu_{i},v_{i}\right)+\overset{p}{\underset{k=1}{\sum }}\beta_{k}\left(\mu_{i},v_{i}\right)x_{ik}+\varepsilon_{i},\;i=1,2,\ldots n$$
where $y_{i}$ is the dependent variable, *x* is the independent variable of the explanatory factor, $\beta_{0}\left(\mu_{i},v_{i}\right)$ represents the intercept at position *i,* $\beta_{k}\left(\mu_{i},v_{i}\right)$ represents the local parameter estimation of the explanatory variable $x_{ik}$ at position *i*, and $\varepsilon_{i}$ is the random error term at point *i.*

The estimated coefficients of GWR are weighted according to the observations and the spatial proximity of a particular point *i*. The parameters can be estimated using the rectangular equation:(10)$$\hat{\beta }\left(\mu ,v\right)={\left(X^{T}W\left(\mu_{i},v_{i}\right)X\right)}^{-1}\;X^{T}W\left(\mu_{i},v_{i}\right)\;Y$$
where $\hat{\beta }\left(\mu ,v\right)$ represents the unbiased estimate of the regression coefficient β, $W\left(\mu_{i},v_{i}\right)$ is the weighting matrix, and *X* and *Y* are the matrices of independent and dependent variables. $W\left(\mu_{i},v_{i}\right)$ ensures that observations close to a specific location have greater weight, expressed using a Gaussian weighted kernel function:(11)$$w_{ij}=exp\left(-\frac{d_{ij}}{b^{2}}\right)\;$$
where $w_{ij}$ represents the weight of observation *j* at position *i*, $d_{ij}$ represents the Euclidean distance between regression point *i* and adjacent observation *j*, and *b* represents the basic width of the kernel function.

Stationarity exists when the variable $x_{ik}$ does not vary with position *i*, and the GWR-based stationarity index is used to estimate spatial stationarity [65]:(12)$$SI=\frac{\beta_{GWR\_iqr}}{2\times GLM\_se}$$
where *SI* is the stationarity index, $\beta_{GWR\_iqr}$ is the standard error interquartile range of the GWR coefficient and GLM_se is the standard error of the global regression analysis. When *SI <* 1, the explanatory variable *y* and the dependent variable *x* achieve spatial stationarity.

AIC can be used to determine the significance of the coefficients to compare relative measures of model performance, the smaller the AIC, the more reliable the model is, and *AICc* represents the limited sample size correction result of the AIC [66].
(13)$$AICc=2nIn\left(\hat{\sigma }\right)+nln\left(2\pi \right)+n\left(\frac{n+tr\left(S\right)}{n-2-tr\left(S\right)}\right)$$
where *n* is the number of samples, $\hat{\sigma }$ is the estimated value of the residual standard deviation, *tr(S*) represents the trajectory of the hat matrix, and when the *AICc* value is lower than three, the model performs better.

To study the driving mechanisms of NDVI in this study we used the local regression method in the GWR Model, taking 14 driving factors related to climate factors, terrain factors and geology and geomorphology as independent variables. In addition, the significance grading result of the NDVI value was used as the dependent variable. Among them, insignificant degradation, insignificant improvement, significant improvement, and extremely significant improvement were assigned as −1, 1, 2, and 3 respectively to represent the change in NDVI. Thus, the estimation coefficients of different factors on NDVI at more than 20,000 sampling points were obtained. To control the accuracy of the estimated coefficients, the outliers of the estimated coefficients of each driving factor were deleted, and the distribution map of the estimated coefficients was obtained through interpolation.

## 3. Results

### 3.1. Spatiotemporal Characteristics of NDVI Trends

#### 3.1.1. Temporal Characteristics of NDVI Trends

The average value of NDVI in the whole region from 2000 to 2020 generally displayed an upward trend (Figure 3). The NDVI value changed from 0.40 to 0.51 over 21 years, roughly increasing by about 0.11, but the R^2^ value was smaller, at 0.486. The variation in the mean value of NDVI had a small amount of fluctuation, and the fluctuation period was about 3 years, reaching a minimum value in 2012 and a maximum value in 2018. Affected by the Wenchuan earthquake and secondary geo-hazards, the NDVI value of the whole region was in a continuous downward trend from 2009 to 2012, and gradually increased in the following years. Although the NDVI of the whole region was generally on the rise, the variation characteristics of NDVI in each subregion were different. Among these, the NDVI value in DA generally showed a downward trend, with a serious and continuous decline from 2007 to 2012 and reached the lowest value of 0.3136 in 2012. In the past 21 years, NDVI values in other regions have been rising, but due to the influence of temperature, the NDVI values in the whole region decreased significantly in 2012.

To obtain the correlation of the NDVI trends and average values between different zones, we conducted a Pearson analysis on the variations in the average NDVI values between different zones from 2000 to 2020 (Figure 4). Except for DA, the correlation coefficient between NDVI trends in the whole area and each area reached more than 0.85. Affected by the surrounding crustal movement, the variation in the NDVI value in DA was unstable, and the correlation coefficient with other areas was low, ranging from 0.27 to 0.59. Due to the low altitude, suitable temperature and sufficient precipitation, the correlation coefficients of BSMS and DXLXXL with the whole region reached 0.96 and 0.97, respectively, which can better represent the NDVI trends in the whole region.

#### 3.1.2. Spatial Characteristic of NDVI Trends

There were few human activities in the SCOCGPNP, and construction land accounted for only 0.6%, so the distribution of NDVI was mainly affected by natural factors. In the study area, the high NDVI values were mainly distributed in the BS, the eastern part of BSMS, the eastern part of DXL and other places with lower altitudes (Figure 5a). Affected by high altitude, the western region has barren vegetation, covered glaciers and low annual precipitation, with an annual precipitation level below 1000 mm all year. The average temperature in some areas was lower than 0 °C, and the basic conditions for vegetation growth were not met. Therefore, the grassland area accounted for more and the woodland accounted for less growth. The overall value of NDVI showed a decreasing trend from northwest to southeast.

To analyze the spatial improvement and degradation of NDVI values in the study area, in this study we calculated the NDVI variability (Figure 5b), and then divided the calculated NDVI variability per pixel into four levels according to the significance classification standards (Figure 5c) to obtain the spatial heterogeneity of the NDVI trends over 21 years. During the period from 2000 to 2020, the NDVI trends showed an overall trend of improvement, and a vegetation coverage rate of 94% of the study area also showed a trend of improvement with an annual growth rate of about 4.7%/yr. The proportion of non-significantly degraded areas was 6%, mainly distributed in DA. The land-use types consisted mainly of shrubs and grasslands, and these were close to the Wenchuan earthquake-generating fault zone, which is prone to secondary geo-hazards. The extremely significant improvement area accounted for 47% of the whole area, mainly distributed in the north of BSMS, the southwest of DXLXXL, and the XXL area, which had high NDVI values, abundant precipitation, and high temperature. The significant improvement area accounted for 47% of the whole area, with scattered areas of non-significant improvement and extremely significant improvement, and most of the land-use types were classed as forest land. The SCOCGPNP, from northwest to southeast, is located in a transition zone from a subtropical zone to a warm temperate zone and a transition zone from the Qinghai Tibet Plateau to the Sichuan Basin. Therefore, the NDVI trends can be roughly divided into the northwest region and southeast region, and the temperature in the southwest region is low all year, which is not conducive to the growth of vegetation, whereas the temperature in the southeast region is high all year, especially in the south.

#### 3.1.3. Verification of NDVI Trends

To test the accuracy of the NDVI trend analysis, we selected five verification areas in different nature reserves for accuracy verification (Figure 5c and Figure A2). Figures show that the Landsat Image of each verification area is consistent with the change trend of vegetation coverage calculated by MODIS NDVI data. In addition, due to the significant degradation of DA, the images of 2000, 2007, 2008, 2009, and 2020 were selected to verify the vegetation changes before and after the Wenchuan earthquake of 12 May 2008. The selected images are Landsat 5 images synthesized by bands 3, 2, and 1 in 2000, 2007, 2008, 2009 and Landsat 8 images synthesized by bands 4, 3, and 2 in 2020 (https://earthengine.google.com/, accessed on 10 May 2022). In DA, the NDVI value changed greatly from 2007 to 2009 (Figure 6). Due to the influence of the Wenchuan earthquake, the relatively strong vibration around the fault caused varying degrees of vegetation damage [67]. Secondary disasters such as landslides and collapses caused by the earthquake in this area gradually transformed the forest land into grassland or bare land [68,69].

### 3.2. NDVI Driver Analysis

#### 3.2.1. NDVI Climate Driver Analysis

From 2000 to 2020, the average value of NDVI, annual precipitation, and annual temperature all showed an upward trend (Figure 7). The average range of NDVI was mainly between 0.4 and 0.5, showing a downward trend from 2009 to 2012, but there was a slight increase in the past 21 years. The average annual temperature range was mainly between 5.6 °C and 6.6 °C, reaching the highest value in 2006 and 2016 and the lowest value in 2000. The annual precipitation ranged from 700 mm to 900 mm, reaching the highest value in 2013 and the lowest value in 2006. The precipitation fluctuated greatly from 2000 to 2006, and the range of fluctuation in temperature was similar to that of NDVI.

Temperature and precipitation are two factors that directly affect the spatial distribution of vegetation. In this study, we calculated the partial correlation coefficient between NDVI, precipitation and temperature from 2000 to 2020 pixel by pixel. In general, there was a strong positive correlation between temperature and precipitation and NDVI. The partial correlation coefficient between temperature and NDVI was greater than zero in more than 90% of the regions (Figure 8a), indicating that temperature and NDVI had a basically positive correlation. The high values of the partial correlation coefficient between temperature and NDVI were mainly distributed in the BSMS and DXLXXL regions, which have a low altitude and high temperature. The partial correlation coefficient between precipitation and NDVI was greater than zero in more than 30% of the regions (Figure 8b). The places with a high correlation between precipitation and NDVI were mainly distributed near the boundaries of three geomorphological divisions in MS, the Wolong Nature Reserve, and the eastern part of XXL.

In this study, a *t*-test with a confidence level of 0.05 was carried out on the analysis results. In the partial correlation *t*-test between NDVI and temperature and precipitation the proportions of results passing the 0.05 confidence test were 51.02% and 9.14%, respectively (Figure 8c,d). After the complex correlation analysis between precipitation and temperature (Figure 8e) the analysis results were tested with a confidence level of 0.05 (Figure 8f), and the value of NDVI in 34.22% of the region was driven by climate factors, which was mainly distributed in the north and south of Southwest Subalpine and Middle Mountain. The average temperature in this region was approximately more than 12 °C, the average annual precipitation was more than 1000 mm, and the variability of precipitation and temperature was large, providing sufficient water and light for vegetation.

According to certain climate-driving factor zoning principles, the climate-driving factors of the whole region were divided into four types: temperature-driven, precipitation-driven, temperature- and precipitation-driven, and other driving modes (Figure 9). The precipitation-driven area accounted for 1.32% of the whole area, which was distributed in the middle of the Xuebaoding Nature Reserve and the Wolong Nature Reserve. The area driven by temperature accounted for 27.46% of the whole region, which was distributed in the west of MS and DXLXXL. The area driven by other driving modes accounted for 65.77% of the whole region; the elevation was higher in this region, and the land-use type was mainly bare land or grassland. Most climate-driven regions had higher NDVI values, higher temperatures, and higher average annual precipitation, which had a significant positive effect on NDVI. Other driving mode areas were generally located at higher altitudes, with lower NDVI values. In these areas, the vegetation was dominated by grasses, lichens, and mosses; the air pressure was low and the carbon dioxide content was much lower, so the driving effects of precipitation and temperature were weak. The driving mechanism in this region is very complex, so it is necessary to use the GWR model to identify the comprehensive driving mechanism of NDVI changes including climate factors based on the analysis of the correlation between climate and NDVI changes.

#### 3.2.2. NDVI Driver Analysis Based on the GWR Model

In this study we used the inverse distance weight to interpolate the spatial distribution relationship of the continuous estimated coefficients between each driving factor and NDVI (Figure 10). The average values of the estimated coefficients of each factor were sorted as follows: TEM_MN > TEM_BT > PRE_MN > PRE_BT > DEN_RIVER > PGA > LUCC > ED_FAULT > NDVI2000 > ED_ROAD > Aspect > Elevation > ED_BLAND > Slope. This indicates that natural factors occupied a dominant position, the variability-driving effect of climate factors was strong, and the correlation between Euclidean distance from built-up land and slope was low.

(1)Driving Effect of Climate Factors

The driving effect of climate factors on NDVI is obvious, and the climate in most areas played a driving role. High temperature and more precipitation were conducive to the growth of vegetation. The positive-driving area of the average temperature accounted for 91.95%, mainly in DA. The negative-driving effect of the area was small, and the absolute value of the estimated coefficient was only 0.66. In areas with higher altitudes, due to the limitation of the growth environment, temperature mostly drove NDVI negatively, and had no obvious promoting effect on vegetation growth. The estimation coefficient of temperature variability was mainly positive-driven and concentrated in MS, whereas the region with a large increase in temperature had less of a negative-driving effect on NDVI.

The positive-driving effect of precipitation-related factors on NDVI was smaller than that of temperature-related factors. The positive-driving area accounted for 25.91% of the estimated coefficient of the average precipitation and was mainly located in DXLXXL. The negative-driving area accounted for a large proportion, but the absolute value of the estimation coefficient was small, and was concentrated in MS.

(2)Driving Effect of Geomorphological Factors

In areas where the altitude was higher, the reduction in temperature and oxygen was not conducive to the growth of vegetation, and the driving effect of slope and NDVI was weak. The negative values of altitude estimation coefficients were mainly concentrated in high-altitude mountainous areas in BSMS and MS. The driving effect of slope on NDVI values was generally small, and the negative-driving area was mainly concentrated in Wolong Nature Reserve where (compared with the surrounding areas) the slope was relatively large. Most of the aspect estimation coefficients were positive, accounting for 87.44%, mainly concentrated in the middle of the study area.

(3)Driving Effect of Geological Activities Factors

In this study, we selected the Euclidean distance from the fault and the peak acceleration of the ground motion as the driving factors related to the geological and geomorphology, and the driving effects of the two on the NDVI were mainly positive. The positive-driving area in terms of the Euclidean distance from the fault accounted for 46.46%. It was mainly concentrated in the large area near the Longmenshan fault zone, which is sensitive to the activity of the fault zone. The negative-driving area accounted for 53.54%, which was mainly concentrated in the south and north of the study area. The positive-driving area of seismic peak acceleration accounted for 64.74%. It was mainly concentrated in DA, which was seriously affected by the secondary disaster of the earthquake. In the northeast of BSMS, the Xuebaoding nature reserve, and the southwest of DXL, the stratum was not active, so its driving effect was weak.

(4)Driving Effect of Human Activity

Human activities have a certain blocking effect on vegetation, in which the high density of roads is not conducive to vegetation growth, and the distance from construction land also has a negative impact on vegetation growth. The negative-driving area of road density exceeded 60%, mainly concentrated in the north of BSMS and DA. The positive value of Euclidean distance from built-up land accounted for 52.52%, and it was concentrated in Da and DXL. The area was close to the construction land, and the NDVI value was low, reflecting the restrictive effect of human activities on vegetation growth.

(5)Driving Effect of Other Factors

The distribution of rivers and the change in land-use types can represent the distribution of NDVI values, and the NDVI value in the initial year can indicate whether the NDVI values in some areas have reached saturation. The river density mainly had a negative impact on the NDVI trend, accounting for 76.44%, mainly in the south of DXL. The change in land-use types reflected the changes in vegetation from a dynamic point of view. The land-use change index was highly consistent with the change trends of NDVI, and the positive-driving area accounted for 85.47%, mainly concentrated in DA and DXL. The negative distribution of the estimated coefficient of NDVI in 2000 was closely related to the elevation. In the southeast of the study area and other low-altitude areas, the positive-driving area of NDVI value in 2000 accounted for 52.43%. The regions with higher altitude were limited by the environment, and the NDVI value reached a certain degree of saturation and no longer had growth potential.

(6)Gradient Variation in Each Driving Factor

To study the gradient variation in each driving factor in SCOCGPNP from northeast to southwest, the study area was divided into five areas, and the numerical distribution of each driving factor estimation coefficient in each area was counted to obtain the histogram shown in Figure A3. The estimated coefficients of the average values of temperature and precipitation tended to increase from northeast to southwest, and most of the estimated coefficients of the average values of temperature were positively correlated, especially in DXL and DXLXXL, most of which were concentrated between 2 and 3. Among the topographic factors, the elevation of BSMS in the Sichuan Basin mainly had a negative correlation with the NDVI value, and the estimation coefficient was distributed between −4 and 1; whereas the estimation coefficient in DXLXXL was mainly concentrated between zero and three, with an obvious positive-driving effect. The road density had a great impact on the NDVI trends in DA, with the slope around the road being large and close to the fault zone. In DXLXXL, the negative impact of river density on vegetation was more obvious, and the area had high annual precipitation and a large river flow. The Euclidean distance to construction land had a great impact on the vegetation growth in DA, as this area is adjacent to the urban area of Chengdu in the southeast and close to the urban area of Wenchuan in the northwest and is thus greatly affected by the development of urbanization. The low-altitude areas of DA and DXLXXL were greatly affected by human activities, and the positive-driving effect of the land-use change index on the change in the NDVI value was more obvious.

## 4. Discussion

### 4.1. Analysis of NDVI Trends

In the past two decades, mechanisms such as natural forest protection, ecological compensation, ecological protection, and regional sustainable development in the study area have all played positive roles in improving the vegetation environment. The Chinese government implemented the project of returning farmland to forest in 2002, and the area of farmland being returned to forest in Sichuan Province reached 77.6 × 10^4^ hm^2^ [70,71]. From 2000 to 2020, the NDVI trends, precipitation, and temperature showed upward trends. The years with a decreasing trend in NDVI change were similar to the years with low values of temperature. With the decreases in temperature in the study area in 2008, 2012, 2014, and 2018, the NDVI value was affected to varying degrees. Affected by the 2008 Wenchuan earthquake and its secondary disasters, the NDVI value in DA showed a continuous downward trend from 2009 to 2012. As the precipitation in the study area in 2011 was lower than that in previous years [72] due to drought events, the NDVI value of the whole area showed an obvious downward trend in 2012.

The univariate linear regression model was able to accurately and intuitively analyze the past NDVI trends and obtain the variation in vegetation characteristics of SCOCGPNP in time and space. The results showed that the variations in vegetation in the study area were mainly affected by elevation and the fault zone, with spatial heterogeneity. Affected by external forces such as crustal instability near the fault zone, the insignificant degradation areas were relatively concentrated, mainly distributed in DA. This area is located in the core area of the Longmenshan fault zone, with a basic earthquake intensity of VIII and the Wenchuan earthquake intensity of XI. In addition, this area is located in the core area of the rain screen district on the eastern edge of the Qinghai Tibet Plateau, and there are problems relating to remaining mining sites. The stability of slopes in the area has decreased sharply, and collapses, landslides, and debris flows are relatively common [73,74]. Therefore, in the past two decades, the NDVI value in this area has shown no significant degradation. On the contrary, the crust in DXL is not active, the temperature is appropriate, the precipitation is sufficient, and the vegetation growth is relatively stable, so it was found to be in a state of extremely significant improvement. In addition to the geological activities and climatic factors, NDVI is also limited by topographic factors to a certain extent [75], and the boundary between significant improvement and non-significant improvement is roughly the same as the snow line. In high-altitude areas, due to perennial snow, thin oxygen, low temperature, insufficient sunshine, and other environmental factors, vegetation generally does not improve significantly.

### 4.2. Driving Force Analysis of NDVI Trends

#### 4.2.1. Climatic Factors

Climate affects vegetation types and spatial distribution [76]. Temperature and precipitation affect plant growth and distribution by affecting effective accumulated temperature and the amount of water available to regulate plant photosynthesis, respiration, and soil organic carbon decomposition. The climate conditions in the study area are diverse. From northwest to southeast, there are plateau temperate humid areas, northern subtropical humid areas, and middle subtropical humid areas [77]. With the increase in temperature, the annual precipitation increases to the level of more than 800 mm, and the water demands of vegetation gradually tend to be saturated, which shows that the sensitivity driven by temperature is higher than that driven by precipitation. In plateau temperate humid regions, such as DA and the Wolong Nature Reserve, due to the limitation of temperature, when NDVI reached saturation, the influence of other driving modes increases. In areas with abundant precipitation, such as the southern DXLXXL and XXL, the area of temperature-driven areas accounts for a larger proportion, because when the precipitation reaches saturation, the increase in temperature enhances the fertilization effect of CO_2_, and the photosynthesis of vegetation is enhanced under high CO_2_ concentration and water stress [78].

Affected by the complex topography of the study area and the transition from a subtropical zone to a warm temperate zone, there was spatial heterogeneity in the distribution of NDVI trends and climate factor regression coefficients obtained via GWR analysis. In the study area, affected by the superposition effect of climate factors, low-altitude areas are positively correlated with NDVI trends, and high-altitude areas are mostly negatively correlated [79]. In BS and the west of MS, the driving effect of average temperature on NDVI trends was mainly negative. In high-altitude, low-temperature, and relatively arid areas affected by the alpine climate the vegetation types are mainly grassland and alpine meadow [80]; the transpiration of vegetation becomes weaker and the improvement effect of temperature rise on vegetation is not significant [81,82]. The negative-driving force of annual precipitation variability on NDVI trends in the study area was mainly distributed in DXL. Because the surface of this area is deeply cut, coupled with frequent seismic activities, the increase in precipitation is accompanied by the occurrence of geo-hazards. At the same time, the presence of complex landforms and a cloudy and rainy environment can also reduce the accuracy of remote sensing results in vegetation monitoring [83].

#### 4.2.2. Geomorphological Factors

Under natural conditions, plant growth and changes are closely related to regional-scale topographic conditions [84] and elevation, cutting depth, and aspect affect soil moisture and the solar radiation distribution. The northwest of the study area is dominated by the Hengduan Mountain Alpine Canyon, which forms a deep elevation difference with the basin in the southeast. The positive and negative-driving effect of elevation on NDVI was about 2500 m, showing vertical zoning. From low altitude to high altitude, the driving effect of elevation on NDVI gradually decreased. The Ya’an area of the subalpine basin in southwestern Sichuan Province and central Yunnan, due to its low altitude, small slope, and sufficient light and heat conditions is conducive to returning farmland to forests. Thanks to the relatively stable geological environment, relatively balanced soil nutrients, and sufficient water conservation, the proportion of vegetation improvement in this area was relatively large [85].

Slope has an important impact on surface runoff and soil properties, and affects the intensity of human activities, so there are differences in vegetation growth conditions in areas with different slopes. In MS and DXL, some areas were affected by slope changes, and soil moisture increased [86], which is beneficial for vegetation to absorb water, and the slope had an obvious positive-driving effect on the NDVI trends. At the same time, ecological projects such as returning farmland to forests and closing mountains for afforestation since 1999 strengthened the protection of forest land [87], and the areas with significant impact were mainly concentrated in low-altitude areas such as the Zhonglongxi-Hongkou area and the Jiudingshan Nature Reserve in Aba Prefecture where there are more transitions of cultivated land to forests, and vegetation coverage increases with the increase in slope.

Aspect indicates the intensity of solar radiation received by the slope and the value and degree of changes in ground water, which affects the sunshine hours and light intensity of vegetation [88]. Compared with topographic factors such as elevation, the driving effect of aspect on NDVI was relatively small. Variations in vegetation were more positively affected by sunlight on sunny slopes. In the southern part of DXL, due to high precipitation, cloud cover and less solar radiation, and relatively wet conditions, the humidity change caused by aspect had little impact on vegetation growth, so the driving force of sunny slopes was small.

#### 4.2.3. Geological Activities Factors

The peak acceleration of ground motion represents the differential movement of crustal fault blocks, and crustal movement is often accompanied by secondary disasters such as landslide and collapse, which affects the stability of the vegetation growth environment [89]. The terrain around DA in the Sichuan Basin is relatively large, and they are mainly considered part of the Longmenshan–Minshan strike-slip thrust seismic zones. When the crustal in situ stress caused by the earthquake exceeds the ultimate strength of the crustal rocks, the rocks fracture and cause surface damage. At the junction of mountains and basins in MS and DXL, the slope is high. Strong earthquakes not only cause a large number of co-seismic landslides, but also exacerbate slope instability for a long time after the earthquake, resulting in drastic changes in land cover in forest or shrubland areas [90], and the vegetation around the Wenchuan earthquake was affected to a certain extent.

Around the fault zone, crustal movement seriously damages the original vegetation and topsoil, forming a large area of secondary bare land [91]. The study area is pushed eastward by the western Qinghai Tibet Plateau [92], resulting in a thrust nappe structure with a sharp difference in altitude. Near the fault zone, the 2008 Wenchuan earthquake triggered rock faults and water and soil losses, which significantly reduced the species richness of woodlands and shrubs, destroyed tree roots, and reduced the density of forest crowns in the Longxi–Hongkou and the Caopo Nature Reserves near Wenchuan in the study area [61]. On the contrary, in the Wolong Nature Reserve, the NDVI showed an improving trend, indicating that after the 2008 Wenchuan earthquake, the vegetation near the fault zone was seriously damaged, resulting in a forest gap [93]. Affected by the amplification effect of slope-secondary geo-hazards after the earthquake, disasters secondary to the Wenchuan earthquake were relatively common in the middle and low mountainous areas. On the contrary, the geological environment of vegetation growth in alpine areas is relatively stable [94]. In XXL, the precipitation level is mostly more than 1000 mm. Affected by the superposition effect of secondary disasters, a large number of collapses and landslides caused by the Wenchuan earthquake provide rich loose solid materials for debris flow activities and also cause a large amount of slope instability and rock mass damage, which greatly reduces the precipitation threshold of debris flow outbreaks in earthquake-stricken areas [95].

#### 4.2.4. Human Activity

The occupation of forest land by road construction for human activities is considered to be an important instigator of habitat fragmentation and biodiversity loss [96]. Road construction has mostly negative effects in areas with large topographic relief, especially in the north of BSMJ and DA. Road construction leads to a change in land use in the places the road passes, which can easily lead to the expansion of cultivated land and the increase in construction land, directly leading to habitat fragmentation and habitat loss, which is the most important and urgent factor threatening the habitat safety of wildlife [97]. In addition, in the south of Wolong Nature Reserve, the G317 national highway passes through areas with deep valleys and large slopes. The construction and use of roads affect the regeneration ability of vegetation to a certain extent, destroying the integrity of forest landscape, and affecting the richness and diversity of vegetation [98].

With the development of urbanization, the expansion of construction land has intensified the impact of human activities on vegetation, and human activities have brought about different degrees of ecosystem degradation. In DA, the area close to the construction land is affected by human activities, which has a high negative effect on NDVI [99], and there are residual problems of mineral development and a decline in the ecological restoration ability of giant panda habitats, which accelerates the fragmentation of protected areas [100]. For example, in the northwest of MS, the construction of the Jiuzhaigou–Mianyang expressway has affected the NDVI value since 2017. In addition, engineering construction activities related to the expansion and development of human society, such as road construction, mining, scenic spots, and water conservancy facilities, are threatening the habitats of giant pandas [101], and the contradiction between ecological environment protection and resource development and utilization is becoming more and more obvious. For example, the opening of the Jiuzhaigou–Chengdu tourism link in the north of the study area and the large-scale local hydropower development have seriously damaged the habitats of giant pandas [102]. However, China entered the scientific development stage of environmental protection in 2002 and then announced a new era of ecological civilization in 2012. During this period, policies for returning farmland to forest and grassland have been initiated as part of ecological immigration policy in extremely important areas of environmental protection [103]. Due to the large distribution of giant pandas in DXL, such as Wanglang, Wolong, and other nature reserves, the ecological migration policy of the nature reserve regulations coordinates the relationship between the community and the ecological environment [104], reducing the adverse impact of human activities. Especially around the Wolong Nature Reserve, construction land has an effect on improving the growth of the NDVI value, affected mainly by tourism, which has transformed the economic structure of the local community [105] and effectively achieved the purpose of sustainable development of the nature reserve.

#### 4.2.5. Other Factors

The distribution of rivers has a regulating effect on the scope of human activities, and the development of agriculture and the construction of water conservancy projects are inseparable from the rivers, which have corresponding effects on the distribution and changes in vegetation. In protected areas such as Wolong in DXL, the positive-driving force of river density on NDVI changes is more obvious; benefiting from the higher elevations in these areas are the development of low-grade rivers, less human disturbance, and better vegetation preservation. However, Ya’an and other areas in DXLXXL were found to have steep slopes, long rainy seasons and concentrated rainfall, fast river velocity, frequent disasters such as floods and debris flows, and severe soil and water losses, resulting in low vegetation development. In addition, human activities around rivers in low-altitude areas are more frequent, and areas close to rivers are prone to the transformation of natural vegetation to farmland, which has a certain inhibitory effect on the improvement of NDVI [106].

Transformation between woodlands, shrubs, swamp wetlands and meadows, and built-up land imply changes in NDVI, resulting in changes in the temporal and spatial patterns of NDVI [58]. In the study area, land-use change mainly has a positive-driving effect on NDVI, which was concentrated in DA. The distribution of plant communities depends on the type of land use, and indirectly affects the vegetation coverage [107]. The transition from the main grassland in the study area to bare land or construction land shows an obvious trend of NDVI degradation. However, the study area was dominated by forest land, and the area of land-use type transformation was small, so the overall driving force of land-use change on NDVI changes was low.

The initial NDVI in the year 2000 can be used to represent the direction of variations, the degree of transition, and the spatial distribution characteristics of NDVI during the 21-year study period. The results show that the initial NDVI and the improvement of NDVI in DA showed an obvious negative correlation because the vegetation cover in this area significantly degraded from 2000 to 2020. The other negative correlations were concentrated in the northwest of BSMS, which was high in elevation and low in temperature, and was covered by ice, snow, and sparse vegetation all year, and the vegetation growth was limited and easily reached extreme values [108]. When the estimated coefficient of NDVI trends for the initial year was at a negative value, this usually indicated a constant or degrading trend in the GWR analysis.

### 4.3. Implications and Limitations

In this research we studied the change trends and driving mechanisms of NDVI in SCOCGPNP using univariate linear regression and the GWR model and obtained effective results, but there were some deficiencies. Due to the limitation of the spatial resolution of the NDVI data, the changes in vegetation growth at the slope scale caused by terrain factors such as surface relief, slope position, and slope may not be fully described. The Savitzky–Golay filtering was regarded as representative of annual data, and the different synthesis methods of data from multiple months in this study area are worth studying. In the GWR model, although the driving analysis considered the impact of land-use changes on NDVI changes, the main body of the study area consisted of forest land and the impact of different forest land types on NDVI changes was not subdivided.

With the continuous improvement of people’s awareness of ecological protection, most of the vegetation in the study area is in a state of improvement through the implementation of measures such as ecological protection zones and artificial afforestation. In key areas of concern, ecological restoration will be gradually achieved through the combination of geological engineering and ecological engineering. At the same time, zoning management should be carried out according to the differences in different natural and human environments. In the Mianyang area of MS, the Ya’an area of DXL and XXL, precipitation is abundant, and the increase in temperature can stimulate the photosynthesis of vegetation which can further improve the vegetation coverage in these areas. To achieve the ecologically sustainable development of GPNP, the government should mobilize surrounding communities to participate in ecological protection projects and build a natural reserve system with national parks as the main body, nature reserves as the foundation, and various natural parks as supplements. During the operation of the national park, a long-term ecological public welfare forest compensation plan and monitoring mechanism should be added, so that residents can actively participate in ecological work. In ecological resource management, attention should be paid to the development of ecotourism, which is an important means of developing natural resource management by the community. At the same time, the construction of GPNP should take the impact of human activities as one of the monitoring indicators of the ecological environment, actively monitor climate change, reduce the risk of geo-hazards, optimize the layout and intensity of human activities, and actively implement measures such as ecological protection and ecological restoration.

## 5. Conclusions

In this study, we not only used the univariate linear model to visualize the trends of vegetation variations in the SCOCGPNP from 2000 to 2020, but also added driving factors such as topography and human activities on the basis of previous studies limited to the driving mechanisms of climate factors in GWR Model, so as to more comprehensively explain the variations in NDVI values. The main conclusions were as follows:(1)During 2000–2020, the NDVI value showed an upward trend as a whole, with a small amount of fluctuation. Affected by the Wenchuan earthquake of 12 May 2008 and its secondary disasters, the NDVI value in DA showed a continuous downward trend from 2009 to 2012. As the precipitation in the study area in 2011 was lower than that in previous years, affected by drought events, the NDVI value of the whole area showed an obvious downward trend in 2012;(2)The NDVI values of the study area showed an overall upward trend from 2000 to 2020, of which 94% of the areas were in an improved state, and the annual growth rate was about 4.7%/yr. The degraded area accounted for 7.94% of the total area, which was mainly concentrated in DA. This area was mainly affected by the Wenchuan earthquake, and the vegetation degradation caused by secondary geo-hazards was more serious;(3)As the study area is located on the geomorphic boundary and climate transition zone, the NDVI trends were mainly affected by the natural environment, in which climate factors were dominant. Moreover, due to the saturation of precipitation in most areas, the driving effect of temperature was more obvious than that of precipitation, mainly concentrated in DXL and DXLXL. The superposition effect of rainfall and topographic factors means the slope had a strong influence on vegetation change, and the areas affected by this were concentrated mainly in BSMS and DA;(4)In the protection of the ecological security patterns of the SCOCGPNP, we should closely monitor regional climate change, prevent, and control geo-hazards, optimize the vegetation growth environment, develop an ecological economy in combination with the current situation of human communities, reduce human interference in the reserve and finally realize sustainable development of the SCOCGPNP.

## Figures and Tables

**Figure 1 ijerph-19-06722-f001:**
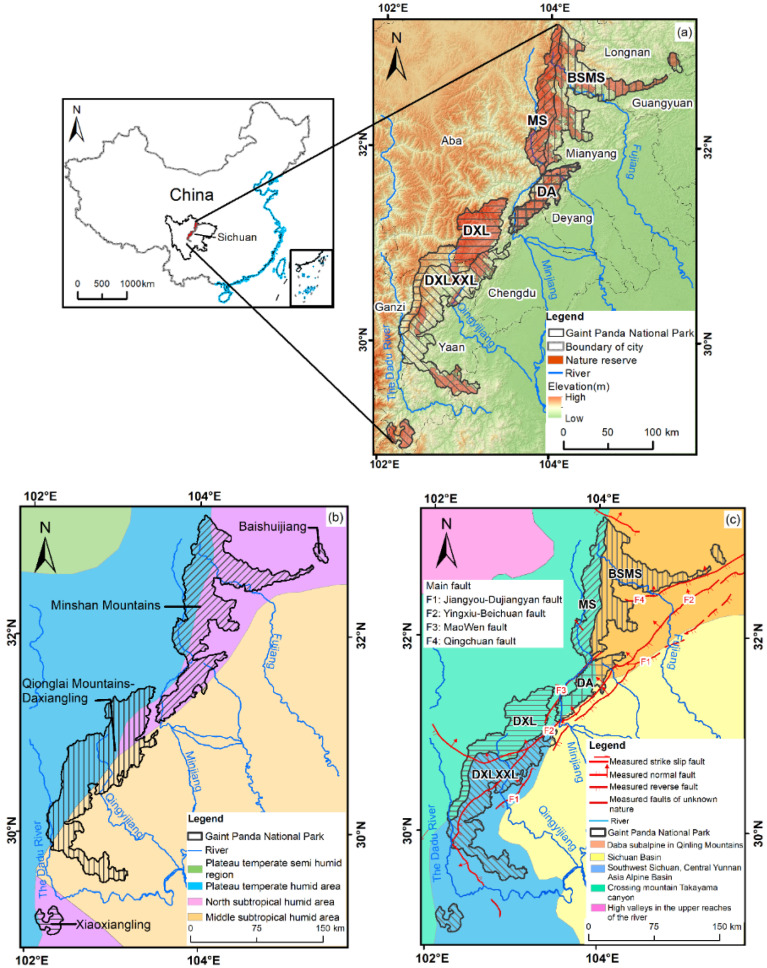
(**a**) Location of the study area. (**b**) Climatic zoning. (**c**) Geomorphological zoning.

**Figure 2 ijerph-19-06722-f002:**
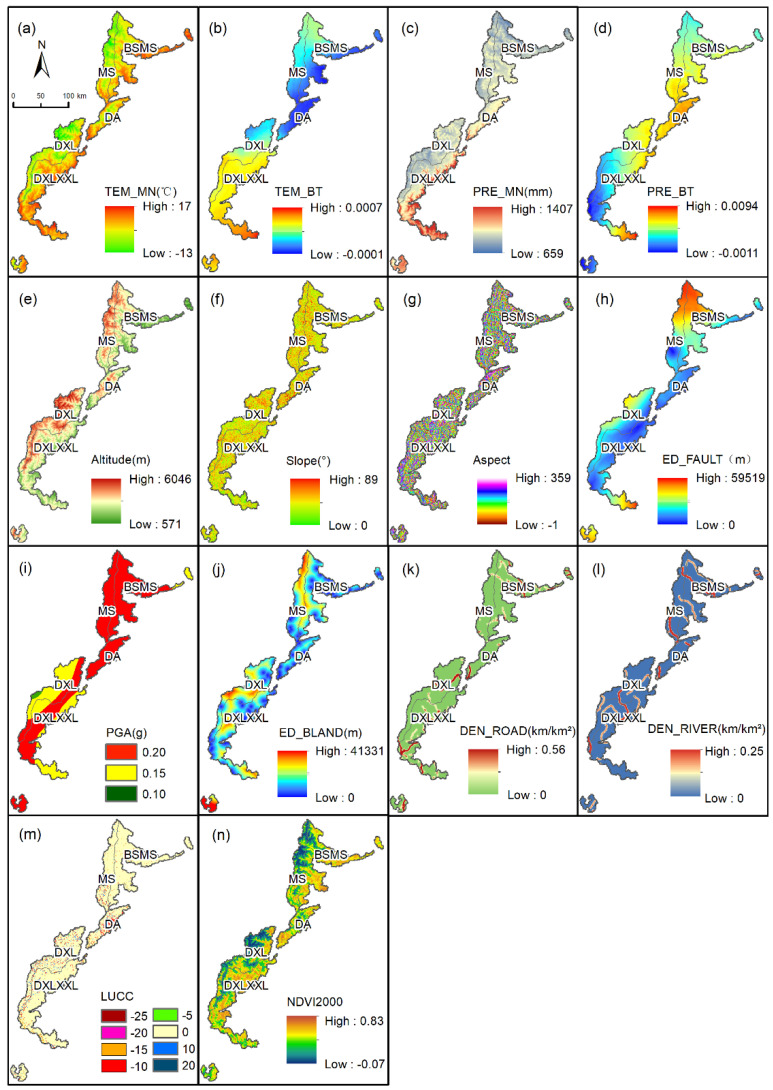
Driving factors used for GWR Model. (**a**) Temperature mean value; (**b**) temperature slope; (**c**) precipitation mean value; (**d**) precipitation slope; (**e**) elevation; (**f**) slope; (**g**) aspect; (**h**) Euclidean distance from fault; (**i**) seismic peak acceleration; (**j**) Euclidean distance from built-up land; (**k**) road density; (**l**) river density; (**m**) land-use change index; (**n**) NDVI in 2000.

**Figure 3 ijerph-19-06722-f003:**
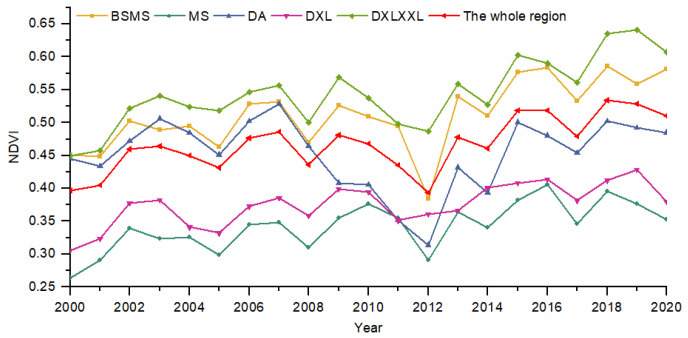
Average value of NDVI in different geomorphological zones from 2000 to 2020.

**Figure 4 ijerph-19-06722-f004:**
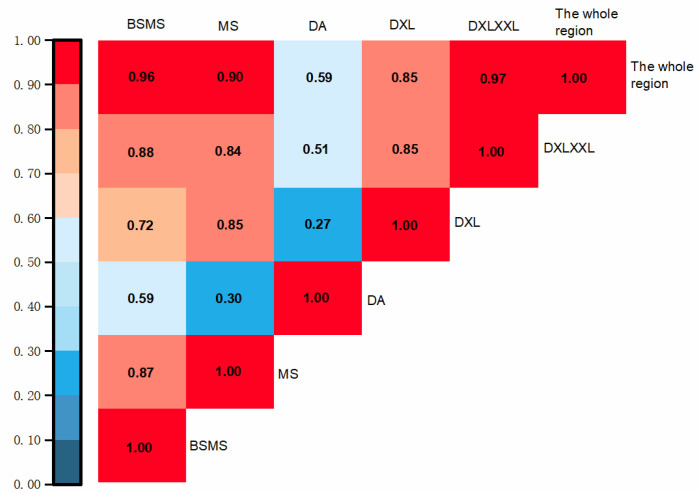
Correlation analysis of NDVI mean values in different geomorphological zones.

**Figure 5 ijerph-19-06722-f005:**
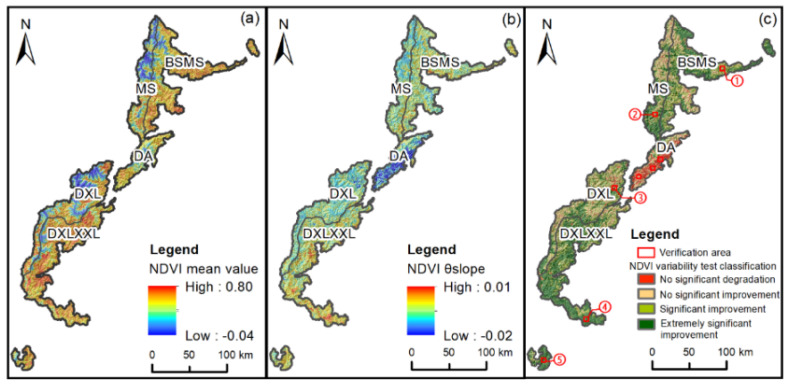
Annual average value of NDVI and its change from 2000 to 2020; (**a**) NDVI average value; (**b**) NDVI $\theta_{slope}$; (**c**) NDVI trends.

**Figure 6 ijerph-19-06722-f006:**
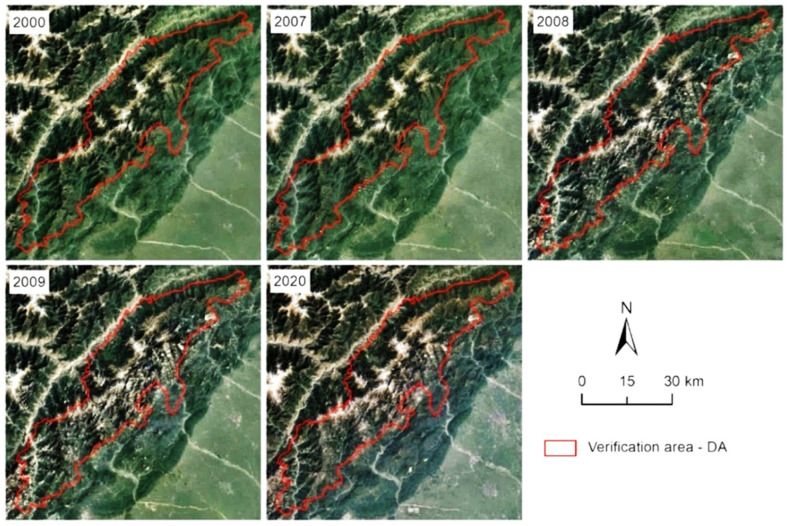
Remote sensing images of DA in 2000, 2007, 2008, 2009, and 2020.

**Figure 7 ijerph-19-06722-f007:**
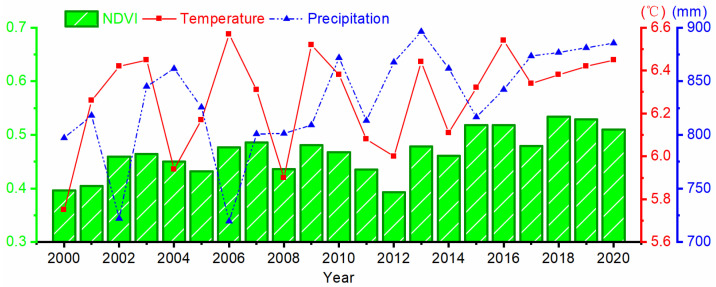
Statistical chart of NDVI, temperature and precipitation from 2000 to 2020.

**Figure 8 ijerph-19-06722-f008:**
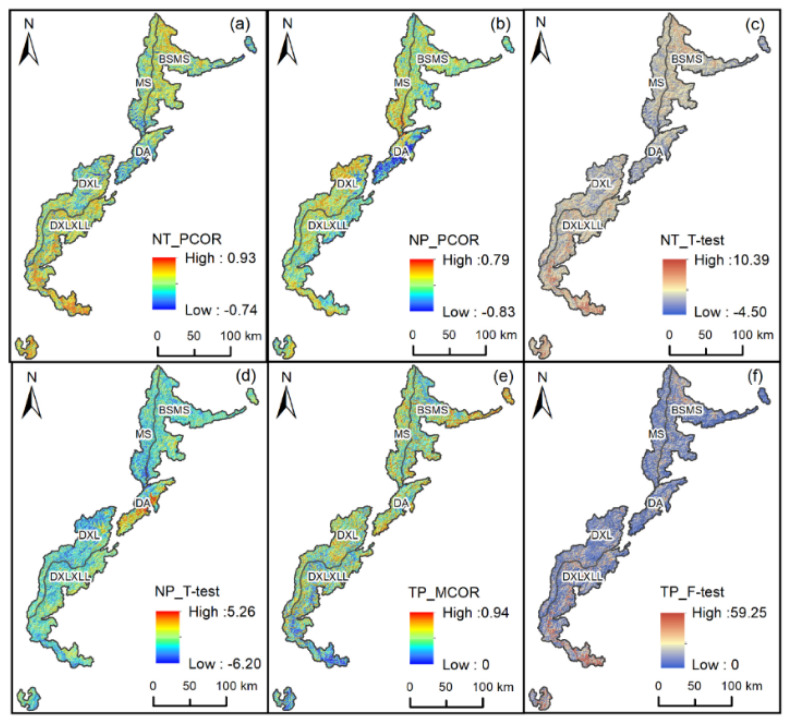
Partial correlation and complex correlation analysis results among NDVI, precipitation and temperature. (**a**) Partial correlation coefficient between temperature and NDVI; (**b**) partial correlation coefficient between precipitation and NDVI; (**c**) partial correlation *t*-test of temperature and NDVI; (**d**) partial correlation *t*-test of precipitation and NDVI; (**e**) multiple correlation coefficient between precipitation and temperature; (**f**) multiple correlation F-test of precipitation and temperature.

**Figure 9 ijerph-19-06722-f009:**
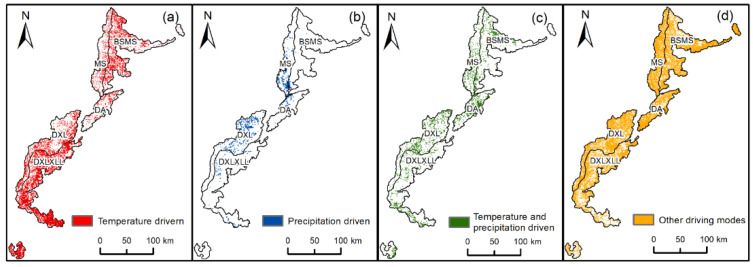
Climate-driven zoning map. (**a**) Temperature-driven; (**b**) precipitation-driven; (**c**) temperature- and precipitation-driven; (**d**) other driving modes.

**Figure 10 ijerph-19-06722-f010:**
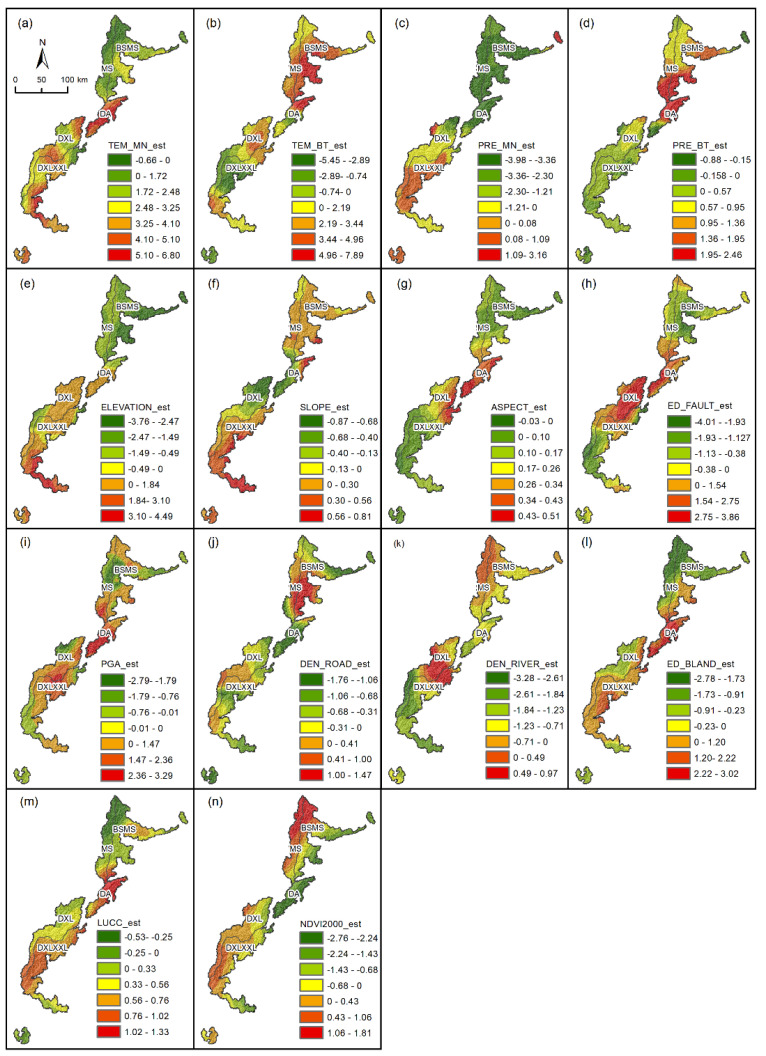
Driving factor estimation coefficient in GWR Model. (**a**) Temperature mean value; (**b**) temperature slope value; (**c**) precipitation mean value; (**d**) precipitation slope value; (**e**) elevation; (**f**) slope; (**g**) aspect; (**h**) Euclidean distance from fault; (**i**) peak ground acceleration; (**j**) road density; (**k**) river density; (**l**) Euclidean distance from built-up land; (**m**) land-use change index; (**n**) NDVI in 2000.

**Table 1 ijerph-19-06722-t001:** Data sources.

Variable Class	Variable Name	Definition and Units	Data Sources	Spatial Resolution
Climatic	TEM_MN	Annual mean precipitation 2000–2020 (mm/yr)	National Qinghai Tibet Plateau scientific data center ^a^	1 km
	PRE_MN	Annual mean temperature 2000–2020 (°C/yr)		1 km
	TEM_BT	*t*-test grading of precipitation trends (OLS) during the growing season 2000–2020		1 km
	PRE_BT	*t*-test grading of temperature trends (OLS) during the growing season 2000–2020		1 km
Geomorphological	ELEVATION	Elevation represents macroscopic geomorphology (m)	Geospatial data cloud ^b^	30 m
	SLOPE	Slope represents groundcutting condition (°)		
	ASPECT	Aspect represents ground orientation		
Geological activities	ED_FAULT	Euclidean distance from fault (m)	China Geological Survey ^c^	Vector
	PGA	Peak ground acceleration (g)	China earthquake administration ^d^	Vector
Human activity	ED_BLAND	Euclidean distance from built-up land (m)	Data Sharing and Service Portal ^e^	30 m
	DEN_ROAD	Road density (km/km^2^)	National Geomatics Center of China ^f^	Vector
Others	DEN_RIVER	River density (km/km^2^)	National Geomatics Center of China ^f^	Vector
	LUCC	Land-use change index	European Space Agency ^g^	300 m

^a^ http://data.tpdc.ac.cn/zh-hans/ (accessed on 25 February 2021); ^b^ http://www.gscloud.cn/ (accessed on 5 May 2021); ^c^ https://www.cgs.gov.cn/ (accessed on 20 May 2021); ^d^ https://www.cea.gov.cn/ (accessed on 20 May 2021); ^e^ http://data.casearth.cn/en/ (accessed on 16 April 2021); ^f^ http://www.ngcc.cn/ngcc/ (accessed on 18 April 2021); ^g^ https://www.esa.int/ (accessed on 25 April 2021).

**Table 2 ijerph-19-06722-t002:** Significance classification criteria.

$\theta_{slope}\;$	*p*	Significance Test Classification
$\theta_{slope}$< 0	*p* ≤ 0.01	Extremely significant degradation
	0.01 < *p* ≤ 0.05	Significant degradation
	*p* > 0.05	Insignificant degradation
$\theta_{slope}$> 0	*p* > 0.05	No significant improvement
	0.01 < *p* ≤ 0.05	Significant improvement
	*p* ≤ 0.01	Extremely significant improvement

**Table 3 ijerph-19-06722-t003:** Zoning criteria for climate-driving factors.

Driven Type	Zoning Criteria		
	r NDVI P, T *^a^	r NDVI T, P *^b^	r NDVI P, T *^c^
Precipitation-driven	*t* ≥ *t*_0.05_ *^1^		F ≥ F_0.01_ *^2^
Temperature-driven		*t* ≥ *t*_0.05_	F ≥ F_0.01_
Temperature- and precipitation-driven	*t* ≥ *t*_0.05_	*t* ≥ *t*_0.05_	F ≥ F_0.01_
Other driving modes			F ≤ F_0.01_

*a Partial correlation coefficient between NDVI and temperature. *b Partial correlation coefficient between NDVI and precipitation. *c Multiple correlation coefficient between precipitation and temperature. *^1^ Confidence *t*-test significance level of 0.05. *^2^ Confidence F-test significance level of 0.01.

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
