# Peer review of "Spatial-Temporal Evolution and Driving Forces of NDVI in China’s Giant Panda National Park"

_ijerph, 2022, doi:10.3390/ijerph19116722_

Round 1

Reviewer 1 Report

This article is about one of the largest and most important areas in the world to maintain the ecosystem of a significative animal like the Giant Panda. In this sense, the use of remote sensing tools is a good strategy and as authors reflected in their article, NDVI could be a good tool to assess ecological security. Moreover, the conclusions are clear and the recommendations given to protect the area from human activities are in the line of the global strategies to keep ecosystems services.

The use of MODIS is justified by the large area studied. I miss the review of other similar works than in regions of North-America and in Mediterranean regions. The literature used is of high interest.

In order to facilitate the reading of the article, I suggest that some words that can be found at the end of a line, would be separate in a better way for reding. For instance, in page two: “gov-ernment” may be easy to read if the separation is “go-verment” or “gover-ment”. This is just a minor comment. Probably you can check similar words cut at the end in other lines of the text, i.e. “signif-icant”, would be better “signi-ficant”.

What I found difficult, was to see some of the figures, with a small size like figure 5. I think this is a key figure to understand the work done.

And finally, as a suggestion. I think that some of the factors used would be redundant, in the way that for instance, precipitation and temperature are in part, presented in the results of the NDVI due to their clear effect in the vegetation. However, I understand the use of them separately as climate factors influencing in the area. Probably, some comments would be included in the discussion.

I agree that human pressure and NDVI are closely related and this would be treated in depth in other articles although this presented is a good starting point.

Reviewer 2 Report

Dear authors,

I found the manuscript interesting and with sound results. I will be very pleased to see it published in its final form in the journal IJERPH.

My comments are mainly concern with the way work is presented and less with the contents. Nevertheless, I would like to see them considered in a final version of the manuscript.

#1 Figures and tables captions

The legends (captions) of figures and tables should be revised. I suggest a detailed revision of all the captions, these must be the most complete as possible, because figures should be legible without the need to see the main text. For instance, the tables and figures captions do not refer the study area or the study context.

#2 “Geomorphic”

I suggest to use “geomorphological” (already used in some parts of the text) instead of “geomorphic” (for instance in figure 1 and table 1, but also in other parts of the main text). 

#3 Pictures

It would be important to have photographs (some examples at least) of the sites studied and described in the manuscript.

Regards

Reviewer 3 Report

Dear Authors,

your article provides with valuable insights into the understanding of the environmental state and dynamics of an internationally- and nationally-important protected area. The study is based on reasonable and in-depth approach, and the outcomes are communicated appropriately. I appreciate the well-fixed structure, numerous illustrations, and the lengthy list of references. Of course, there is something to improve – see my recommendations below.

  • Title: please, simplify and avoid abbreviations.
  • Abstract: please, shorten.
  • Key words: please, avoid the words from the title.
  • Figure 1: what are the sources?
  • Speaking about geology, you write chiefly about seismicity and landslides. Calling these "geological environment" is too ambitious and may cause some confusion. For instance, you write in one place about "the geological environment of vegetation growth". This expression is confusing because one would expect something like lithological controls on vegetation type. So, I advise you to think about the related terminological changes for better clarity.
  • Speaking of human activities, what about environmental pollution from urban areas and tourism activities?
  • Subsection 4.2.3: the word "landslides" is duplicated occasionally on the Line 9 of this subsection.
  • Certain linguistic correction is necessary.
  • ALL abbreviations should be explained in the first place of their use (I recommend to avoid them in the abstract at all).
  • I also advise to make the section "Conclusions" and may be some other parts of the work less technical and more clear.
  • If possible, it would be nice to see some photos of the characterized area, but I do not insist on their inclusion.
